# Ontological Beliefs and Hunter–Gatherer Ritual Landscapes: Native Californian Examples

## David S. Whitley

Rock Art Research Institute, University of the Witwatersrand, Johannesburg 2017, South Africa; huitli53@gmail.com

**Abstract:** Landscapes are socially produced and reproduced spaces. This is easily recognizable for large-scale urban groups with built environments that dominate living places. But it also pertains to all types of societies and cultures, even small-scale hunter–gatherers, once the ontological beliefs structuring landscape perception and use are acknowledged. The foragers of south–central and southern California and the Great Basin illustrate this fact. They maintained a widely shared ontological perspective supported by a fundamental cognitive postulate. This is that supernatural power, the principle causative agent in the universe, was differentially distributed among individuals and places. The distribution of power, revealed by certain geomorphological features and natural events, structured their perceptions of landscape. These perceptions were expressed in ritual and symbolism, including petroglyphs and pictographs as durable manifestations of ceremonies on the landscape. The ontological relationship between power and landscape explains a longstanding question in hunter–gatherer archaeology: Why were rock writing sites created at specific locations? It also explains another equally significant but rarely considered and related problem: Why do some localities have massive quantities of rock writings that dwarf most other sites? The landscape symbolism of and the placement of sites by Native Californian and Great Basin tribes is explained by reference to their shared ontological beliefs, illustrating how they structured their ritual practices and archaeological record.

**Keywords:** archaeology; landscape; ritual; religion; rock art

## 1. Introduction

Humans have been hunter–gatherers for the vast majority of our existence. Foraging as a lifeway arguably is the single most successful ecological adaptation ever inasmuch as it allowed our species to colonize the entire habitable world, from the arctic through the mid-latitudes and tropics to the most arid regions of the world, including many places where farming could never be successful (cf. Bettinger 2015). Yet despite the historical persistence of foragers in human history, and although there are exceptions, comparatively little effort has been spent investigating their religious beliefs and practices in North America, including their symbolic systems and ontological commitments (Whitley 2014). This partly results from a widespread evolutionary/primitivist bias, which today is often implicit yet nonetheless remains strong: since hunter–gatherers seemingly had relatively simple adaptive, economic and technological systems, their cognitive systems, including ritual and beliefs, likewise *must have* also been simple—if they existed at all. Or so many North Americanists seem to implicitly believe.

Renowned Great Basin ethnographer and anthropological theorist Julian H. Steward (1938, 1941, 1955), for example, characterized Numic-speaking Shoshone lives as principally 'gastric' or 'practical' in nature. They "lived at a bare subsistence level. Their culture was meager in content and simple in structure" (Steward 1938, p. 1). Erminie W. Voegelin, similarly, stated that among the neighboring Tubatulabal, "Symbolism [was] apparently entirely lacking" (Voegelin 1938, p. 59). Yet in both cases, these ethnographers published

information on the most obvious and long-lasting examples of hunter–gatherer symbolism and belief: the pictographs (rock paintings) and petroglyphs (rock engravings)—rock writings—inscribed on their landscapes. Indeed, Steward (1929) even produced the first modern synthesis of Numic (Shoshone and Southern and Northern Paiute) rock writing. This included the petroglyphs of the Coso Range, eastern California, estimated to comprise as much as a million or more individual motifs. That Steward at once analyzed one of the world's largest and most dramatic expressions of hunter–gatherer ritual, belief and symbolism, yet still insisted that their lifeways primarily emphasized the simple and mundane, is a testament to the strength of preconceived biases about these people. The current emphasis on behavioral ecology in hunter–gatherer studies continues, even if inadvertently, this intellectual tradition by foregrounding adaptation and diet as (somehow) the central features of forager lives. While at certain level it is true that we are what we eat, this reductionist view ignores the full measure of Native American lives. Archaeologists, as putative stewards of the past, owe more to the archaeological record than continuing to exclusively promote a mind-blind view of the Native American past.

I describe in the following, accordingly, aspects of the religious beliefs and ritually produced symbolism—rock writing—in the context of shared ontological beliefs about landscape, for southern and south–central Native Californian and Great Basin tribes[1]: Numic speakers of eastern California and the Great Basin; the Hokan-speaking Chumash of the Santa Barbara coast and adjoining interior region; the Penutian-speaking Yokuts of the San Joaquin Valley and southern Sierra Nevada; and the Takic speakers of southernmost California. My analysis pivots on a shared cognitive postulate: their concept of supernatural power, where it resided, how it was obtained, and its implications for larger cultural patterns and practices. When seen from the perspective of ontological beliefs about power and landscape, it is possible to resolve two longstanding issues in landscape studies of rock writing—why these sites were created at specific locations and why certain localities have unusually large, sometimes massive, clusters of sites and motifs. I start accordingly with a brief discussion of ontology and its implications as well as their central percept, supernatural power.

## 2. Ontology, Landscape and Power

Recent archaeological theory has highlighted the importance of ontological beliefs in understanding pre-contact cultures, with many authors contending that our discipline experienced an analytical 'ontological turn' roughly a decade ago (cf. Alberti 2016). Most of this literature has emphasized the transactional relationship of humans to animals, sometimes referred to as 'non-human people' or 'other-than-human-beings'. Two points are worth noting about this putatively recent research direction and its focus. As noted elsewhere (Whitley 2021), first, the so-called ontological turn in archaeology in fact started a few decades earlier, in global rock art research and its use of symbolic analyses of ethnographic texts (e.g., Lewis-Williams 1981; Whitley 1992; Layton 1992, 2001). These studies provided, among other topics, descriptions of human–animal transformations as so-called therianthropes, in the process outlining an understanding of personhood which involved conflations of human identity, agency and action with spirit beings and spirit animals. They also revealed graphic imagery not simply as a kind of passive text but instead as material 'spirit objects' with their own potency and agency. They further showed that certain affective religious experiences involved embodied physical and emotional reactions that accounted for widely shared practice and beliefs. And they identified landscape features as more than straightforward natural characteristics of the world, instead being potentially invested with supernatural characteristics, implications and meanings (cf. Whitley 2021, 2024).

This substantial record of previous research, second, has likely been ignored by some promoters of a recency in ontological approaches for multiple reasons beyond narrow scholarship, stemming from the general neglect of archaeological rock art studies by non-specialists. Equally contributing, I suggest, has been an overemphasis on transactional

human–animal relationships as seemingly the only ontological concern. While these are important, it needs emphasis that 'ontology' is commonly defined as involving the 'nature of being,' which is to say 'the nature of existence'. Much of the recent archaeological literature might suggest instead that ontology almost exclusively involves a reductionist concern with 'the nature of *beings*' rather than the larger concept of *being*, singular.

This distinction is especially important when considering what we as Westerners label the natural world, beyond animals, for it too is understood in terms distinct from our own metaphysical concepts of it. Relevant to this point, Smith has contended that even our concept of "nature" itself is a cultural construct:

> The idea of the production of nature is indeed paradoxical, to the point of sounding absurd, if judged by the superficial appearance of nature in capitalist society. Nature is generally seen as exactly that which cannot be produced; it is the antithesis of human productive activity...[Yet] when this immediate appearance of nature is placed in historical context, the development of the material landscape presents itself as a process of the production of nature. (Smith 1984, p. 32)

Soja (1989) amplifies Smith's argument in his discussion and analysis of the concept of *spatiality*. This has three related dimensions: socially produced space; the physical space of the natural world; and mental space formed by cognition and representations:

> The presentation of concrete spatiality is always wrapped in the complex and diverse re-presentations of human perception and cognition, without any necessity of direct and determined correspondence between the two. These representations, as semiotic imagery and cognitive mappings, as ideas and ideologies, play a powerful role in shaping the spatiality of social life. (Soja 1989, p. 121)

My concern in this paper involves hunter–gatherer *landscapes*. The concept of 'landscape' (or 'cultural landscape') has traditionally been defined as the humanly-modified natural world, typified by farms, gardens and settlements (e.g., Rubinstein 1989, pp. 28–29). Hunter–gatherers from this perspective did not have (that is, did not create) landscapes because they (supposedly) did not markedly modify the land. Following Smith (1984) and Soja (1989), I define landscape as the cognitive interpretation and understanding of a culture's physical world: its mental construction of "nature" more than its physical alterations to it. The resulting intellectual model is based on their ontological and epistemological beliefs and representations as well as their symbolic system, ritual use of space and traditional ecological knowledge. This landscape can be deciphered, even by a Western scientist, because humans are fundamentally thinking animals; "the symbolic system is highly empirical" (Sahlins 1985, p. xiii); and traditional thought systems are inductively and deductively like our own (Horton 1976, 1982). All humans share a universal cognitive system, operating under the same principles, even if the resulting details of knowledge, belief and metaphysics are not uniform (Menon and Cassaniti 2017; Zerubavel 1999).

In Native California and the Great Basin, ontological beliefs about landscape centered on a widely shared understanding of supernatural power, referred to by Numic speakers as *puha* or *poha*. Released at the creation, power was thought the central causative agent in the world. It was believed to be sentient and was possessed by anything with life or the will to act. This power was differentially distributed both in the universe and in the Middle World of humans, where it was in a state of static equilibrium, with specific individuals serving as the axis or fulcrum for the interactions of power holders. It was different from the Christian concept of grace in that it was morally ambivalent and could be used for positive/good or negative/evil purposes. It was also thought dynamic, with power conceptualized as flowing outwards from a central source (Bean 1975; Miller 1983a; cf. Applegate 1978; Carroll et al. 2004; Liljeblad 1986; Miller 1983b, 1985; Stoffle and Zedeño 2002; Van Vlack 2022; White 1963).

A series of perceptions about supernatural power resulted from these beliefs. Perhaps the most important is that supernatural power was required for success in any activity, religious or secular (Bean 1975; Applegate 1978). Evidence of such success, in a reciprocal

fashion, was necessarily assumed to signal ownership of the associated power (Bean 1975; Whitley and Whitley 2012). Political leaders, good hunters or gamblers, the wealthy and, of course, religious adepts were all inferred to maintain potent levels of the specific kinds of power which afforded them their skills and achievements. Yet power, given its ambivalence, was inherently perilous to those not trained in its use:

> Power is dangerous. Failure to obey [its] commands and instructions will lead to sickness and even death. (Whiting 1950, p. 42)

> Since man is never absolutely certain whether or not anything is a power source until it is tested or reveals itself, he lives in a constantly perilous world fraught with danger. (Bean 1975, p. 26)

The individuals who were trained in the control and exercise of power were shamans or, in the Numic languages, the *puhagunt*, a 'man with power'. These ritual specialists were the "boundary players" who knew the "rules for acquiring, manipulating and controlling power" (Bean 1975, p. 27):

> During rituals, when power is being exercised, past, present, and future may be fused into one continuous whole. A shaman may use power to bring sacred time into the present so that he can interact with beings from that time. (ibid)

All humans, as living beings, had a certain amount of power, but shamans had both significantly more of, as well as the ability to call upon and control, it (Applegate 1978; Miller 1985). They typically received this power during 'dreams' or visionary experiences. These were obtained through true nighttime REM 'power dreams' (which we would label nightmares), isolated/individual vision quests or ritual pilgrimages involving shamans and shaman initiates. Power was typically awarded, after appropriate prayer and training, through the receipt of one or more spirit helpers, usually but not invariably animal in form (Applegate 1978; Hultkrantz 1961; Miller 1983a):

> The intent of this training was the ability to control the supernatural power-energy- force after it was conferred by a spirit in a recurrent dream, which usually was due to the initiative of the spirit, either because it was familiar with the family or because it took a liking to the dreamer. (Miller 1983a, p. 69)

Note that none of the altered state of consciousness/visionary experiences described in the Native Californian and Great Basin ethnographic record for the acquisition of power can be characterized in any sense as ecstatic, contrary to widespread contemporary Western perceptions (Whitley 2009, 2024). These experiences instead served as a kind of initiatory/spiritual test evoking fear if not terror rather than bliss. Taivotsi, a Numic shaman-initiate, for example, was tested during his quest successively by an owl pecking him, a bear grabbing him and a coyote biting him:

> Finally, a big rattlesnake came writhing against him, rattling its tail. Taivotsi had always been afraid of snakes, so he jumped up and ran away. That was his misfortune. If he had stayed on the spot, the snake would have entrusted him with supernatural medicine, for the snake was his spirit. (Hultkrantz 1987, p. 54)

The individual/isolated Numic vision quest, as opposed to the visionary experience per se, is summarized as follows:

> The general rules of the Shoshoni vision quest have been the following. The supplicant, usually male, rides a horse up to the foothills where the rock drawings are—the latter are the foremost places of spirit revelation. At a distance of some 200 yards from the rock with the pictographs he tethers his horse. Then he takes a bath to cleanse himself in the nearest creek or lake. Without moccasins he walks up to the rock ledge just beneath the drawings and makes his camp there. Naked except for a blanket around him, he lies down there under the open sky, waiting for the spirits to appear. Sometimes, I was told, the supplicant directs a prayer to a particular spirit depicted on the rock panel, anxious to receive that very spirit's

power. . .each spirit that blesses its client does so with a special gift related to the spirit's own abilities. (Hultkrantz 1987, pp. 52–53)

Shamans among certain groups, especially the Numic-speaking Southern Paiute, conducted group pilgrimages from which shaman initiates "graduated" as full practitioners. Van Vlack notes that: "In Southern Paiute culture, pilgrimage has always focused on the acquisition of *Puha*, the spiritual transformation of *Puha'gants* (Southern Paiute medicine men) and building relationships and communities" (Van Vlack 2022, p. 33). One description of such a shamans' pilgrimage highlights a key characteristic of the location of these rituals and what occurred at these spots:

There are moments during pilgrimage when pilgrims leave the physical world and enter into a spiritual one. This transition most frequently occurred at petroglyphs and rock paintings. For example, during a pilgrimage to an old volcano in southern Nevada, pilgrims sought the *Puha* from a mountain sheep-head petroglyph by entering into the image through a hole in the rock where the mountain sheep's eye was located. Pilgrims inserted their finger, covered in red paint (*oompi*), into the eye hole. This act transitioned them into the rock and its spiritual dimension. (Van Vlack 2012b, p. 133)

Following a widespread Native American belief about sacred places, rock writing sites were then portals into the supernatural world of power (D. E. Walker 1991; Whitley 1992, 2000; Stoffle et al. 2011b). Entering this world could involve transitioning into or through the rock face. Because the supernatural realm, as noted above, was dangerous, these entries were guarded by supernatural snakes and grizzlies or clashing rocks and boulders (e.g., Blackburn 1975, pp. 197–98; Zigmond 1980, pp. 175, 177, 178–79). Rock writing sites accordingly were "avoided" by non-shamans, in the anthropological sense of this term, even when they were located in villages and despite the fact that non-shamans also used them for prayer, meditation and curing:

pictographs are "out of bounds" for people [i.e., non-shamans]. The paintings may be looked at without danger, but touching them will lead to quick disaster. One who puts his fingers on them and then rubs his eyes will not sleep again but will die in three days. . .it would be necessary for each of us to make an offering to an animal whose representation we chose to see. Otherwise we would see nothing. . .[Informants] told of a non-Indian woman who had come to see the pictographs but made no offering (possibly she was ignorant of the custom!). She heard the growl of a grizzly bear, fled, and never returned. According to one version of the story, she was actually chased by the bear. (Zigmond 1977, p. 79; quote in original)

Rock writing sites were known as *pohaghani*, 'house of supernatural power' (Malouf 1974, p. 81; Hultkrantz 1987; Shimkin n.d.) among the Numic. Paralleling this term, Chumash rock writing sites were called "medicine houses" (Harrington 1926, p. 107), signaling their connection to 'medicine,' which is to say supernatural power, while one of the Yokuts generic names for such sites translates as 'spirit helper place' (Whitley 1992, 2000). These sites were created at the culmination of a shaman's vision quest with the motifs portraying the shaman's spirit helpers and actions in the supernatural (Brooks et al. 1979; Carroll 2007; Driver 1937; Hultkrantz 1987; Lowie 1909; Shimkin n.d.):

In the night, his medicine [spirit helper] speaks to him and counsels him. It may tell him how he ought to paint. (Lowie 1909, p. 224)

Shamans painted their "spirits (*anit*) on rocks to show themselves, to let people see what they had done. The spirit must come first in a dream". (Driver 1937, p. 126; quotation in original)

When ceremony or power seeking is successful at such places, they are selectively marked so future human visitors can more fully understand the purpose of the place. (Stoffle et al. 2011b, p. 11)

Other accounts attribute the creation of rock writing to the actions of the spirit helpers themselves (Hultkrantz 1987; Irwin 1980; Laird 1976, 1984; Stoffle et al. 2011b, 2011c; Voegelin 1938; Zigmond 1977, 1986). This reflects the ontological belief that the actions of a shaman and his helper were indistinguishable (Applegate 1978; Gayton 1948; Laird 1984; Siskin 1983).

## 3. Locating Power

It has been long understood, even if only in general terms, that the locations of rock writing sites were predicated on the distribution of supernatural power (e.g., Whitley 1992; 1994b, p. 7; 1998; Whitley et al. 2004). A recent compilation, analysis and synthesis of relevant ethnographic data on this topic (Whitley n.d.) allows for a much more detailed account of the logic behind the placement of these sites. This was based on ontological beliefs about supernatural power and its manifestation on the landscape. A kind of dialectical relationship then existed between power and the landscape because the land itself was sentient and alive (Stoffle et al. 2022). As noted by Stoffle et al. (2015, p. 100): "landscapes and objects communicate to people how they are intended to be used," indicating that landscape features themselves exercised a kind of agency that humans responded to ritually (Liwosz 2017).

Supernaturally imbued natural features and phenomena, used for the rituals that resulted in rock writing, are outlined below. Note that these distinct kinds of contexts often co-occur, contributing to the perception of even greater supernatural potency for a particular locale, as discussed subsequently.

### 3.1. Geological/Geomorphological Features

Rock writing sites by definition are cultural landscape features located on boulders, cliff faces and cave walls and ceilings; that is, on rocks. Since the sites are 'houses of power,' as the Numic labeled them, it follows that rocks in a general sense were understood as at least potentially associated with the supernatural. Their spiritual significance is in fact emphasized given that entry into the sacred often involved passing into a rock, sometimes through a crack or hole in its face (Blackburn 1975, p. 233; Whitley 1992), or instead when a shaman imbibed a hallucinogen or used his staff to open the passage (e.g., Laird 1974, 1976). Spiritual revelation, furthermore, could in theory happen anywhere, to anyone, at any given time. This was in part because landscapes were thought alive and, as with all living things, some degree of power was then present in all places, even if it was differentially distributed (Van Vlack 2012a; Stoffle et al. 2022). All rocks hence were, practically speaking, appropriate places for revelation and ritual activities. Yet rock writing sites are neither ubiquitous nor uniformly distributed. They are instead typically clustered into certain areas (Whitley et al. 2007; Whitley and Whitley 2012) and often limited to particular kinds of landforms. These correspond to landscape features that have been identified by Native Americans as frequently imbued with unusual amounts of potency.

Significant geological features identified as supernaturally powerful start with mountains, especially (but not only) the highest peak in any groups' tribal territory, which was widely believed to be their creation point (Laird 1976; Hudson and Underhay 1978; Blackburn 1975; Miller 1983a; Fowler 1992; Shipek 1985). Mountains were associated with water (discussed below), with power thought to flow from the peaks, like streams down a mountainside:

> Mountains, which are perceived as powerful living beings, talk with the clouds and encourage them to bring moisture for the land. All high mountains call down moisture, but the highest mountains call down the most. (Stoffle et al. 2015, p. 105)

Mountains of all kinds, but especially those with significant vistas, were then used for rituals, including vision questing, pilgrimages, prayers and cures (Lowie 1909; Kelly 1932; Park 1938; Steward 1941; Stewart 1941; Olofson 1979; Riddell 1978; Miller 1983a; D. E. Walker 1991; Fowler 1992; Arnold and Stoffle 2006; Van Vlack 2012a; Stoffle et al.

2022; Shimkin n.d.). They were the homes of many spirit helpers who were the sources of shamans' power (Kroeber 1907, 1925; Lowie 1909, 1924; Kelly 1932; Stewart 1941; Riddell 1978). Rock writing sites, perhaps surprisingly, nonetheless are only rarely located on mountain tops, despite their spiritual importance. This apparently results from the gender symbolism of the landscape versus that of the petroglyph and pictograph sites specifically: rock writing sites as female-gendered places versus mountaintops as male-gendered (Whitley 1994a, 1998), thereby illustrating the common regional use of symbolic inversion to emphasize the transition into sacred (Blackburn 1975; Applegate 1978), in this case male shamans using a female-gendered place for vision quests. The association of a place or landform with supernatural power alone then does not ensure in all cases that it will contain rock writing.

These sites instead are much more frequently found on two other landforms, the first of which are caves/rockshelters:

> deep caves on slopes are sacred because they shelter life and collect water by seepage while remaining moist and dark like the initial world ... Hence caves are sacred, vital in the flow of power. (Miller 1983a, p. 78)

> [There] are a number of sacred places known to the people as places where power resides ... some of these places (usually caves) were visited by men and women wanting to become doctors [shamans]. But they could also be visited by persons wishing to increase their luck in hunting or gambling, or by persons who wanted children or other special favors...The power found at each of these places was specific to it; i.e., it was localized there. (Fowler 1992, p. 176)

Laird (1976, p. 38) described such sacred caves as "places of power and mystery." In addition to shamanic vision questing, they were also used by non-shamans for prayers, supplication, curing and meditation (Park 1938; Harris 1940; Steward 1941; Malouf 1974; Laird 1976; Zigmond 1977; Riddell 1978; Fowler and Liljeblad 1986; Liljeblad 1986; Kelly and Fowler 1986).

Caves and rockshelters are of course famously renowned for their rock writings, so much so that the sites are sometimes referred to in the popular literature as 'cave paintings'. Rock writing sites are most often associated with true caves, boulder piles creating cave-like environments, and/or rockshelters where these features themselves are common: sandstone formations for caves, typical of the coastal ranges occupied by the Chumash (Figure 1), and granodiorite for boulder piles and overhangs, typical of the Sierra Nevada and many portions of southernmost California (Figure 2).

The second geological context commonly associated with supernatural power is volcanic: cones, domes, lava flows and associated geological features (Arnold and Stoffle 2006; Van Vlack 2012a; Van Vlack et al. 2013; Ruuska 2014; Stoffle et al. 2015):

> Volcanoes have a special place in Southern Paiute epistemology, and Southern Paiute people are strongly culturally attached to volcanic places and events. Volcanic episodes are distinctive moments when *Puha* moves from lower to higher levels of existence and causes the power to accumulate in these areas ... *Puha* moves from the lower portions of the Earth to form hot springs, mountains, volcanic cones, basalt mesas, lava tubes, basalt bombs, and obsidian deposits...Southern Paiute people respect and interact with places of volcanic activity because these places contain powerful forces, spiritual beings who can balance human society at local, regional, and world levels. As one Southern Paiute elder said, "Volcanoes are sacred mountains. The old people knew it was alive, like the mother earth is alive. We have a song about the rocks shooting out of a volcano near home". (Van Vlack 2012a, p. 224; quotation in original)

> Religious leaders pass through initiation stages on volcanoes, which often provide visions, songs, and even physical objects such as obsidian, paint, crystals, and turquoise. The pilgrimage paths to volcanoes are repeatedly traversed. Offerings

mark key places along the trail, thus building a permanent record of devotional behavior. (Stoffle et al. 2015, p. 100)

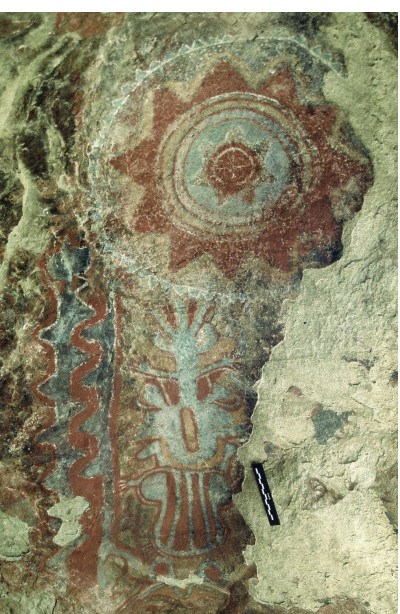

**Figure 1.** Small caves are relatively common in the Californian coastal ranges, characterized by sandstone and other sedimentary formations. This Chumash example, from Pleito Creek Cave on Rancho San Emigdio, is located near the San Andreas rift zone, discussed subsequently. The central figure is an anthropomorph showing a torso successively "exploding" out of the lower shoulder area, up to the mandala like motif at the top. (Note the portrayal of the hands and fingers). A zigzag/rattlesnake, to the left, illustrates the belief that entry into the sacred required crossing a dangerous supernatural snake. (Scale in centimeters; photo by D.S. Whitley).

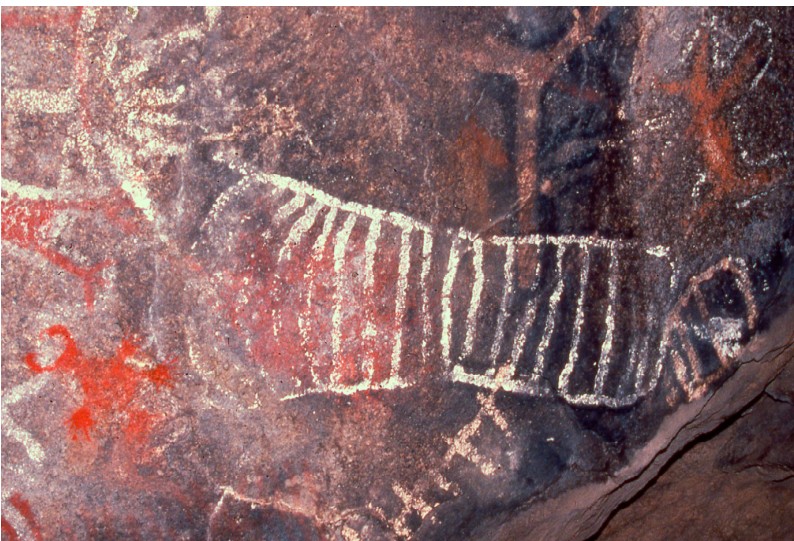

**Figure 2.** Rockshelters, many created by boulder piles, are common in those parts of California where bedrock is predominantly granodiorite: the southern Sierra Nevada and southern part of the state especially. This Yokuts example, from Rocky Hill, has been described by a descendant of the painters as portraying a 'bridge to the supernatural'. The red horned figure at bottom left is an evil spirit that is sometimes encountered in the supernatural. This panel appears to have been painted in the first half of the twentieth century. (Length of ladder-like 'bridge:' approximately one meter. Photo by D.S. Whitley).

Volcanoes were understood as places where the earth is renewed and reborn (Arnold and Stoffle 2006), thereby serving as conduits for the emergence of supernatural power into the Middle World of humans and giving such places a special kind of potency. Volcanic landscapes are particularly common in eastern California and the Great Basin. Although these landforms have been largely geophysically dormant in the recent past, a number of eruptions have occurred in the last few thousand years (e.g., Klinger 2001, p. A2; Fierstein and Hildreth 2017; Bevilacqua et al. 2018)—recent enough for these impressive events to have become part of shared cultural memories, as the ethnographic accounts attest. Petroglyphs in particular are commonly found on volcanic (basalt) outcrops in this region, where their rock-varnished surfaces provide an ideal canvas for engravings (Figure 3).

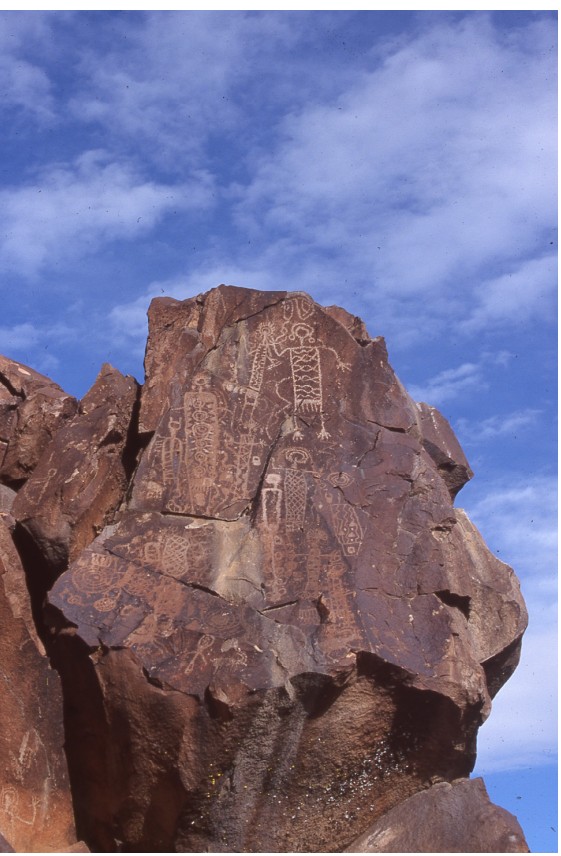

**Figure 3.** Rock-varnished basalt cliff faces and boulders are an ideal medium for engravings. These contexts are common in the California deserts and Great Basin where volcanic formations are likewise typical. This panel, from the Coso Range, California, shows a series of 'patterned-body anthropomorphs:' shamans wearing the ritual shirts that they painted with their signs of power. Variations in the lightness of the motifs indicate different ages for the individual motifs and, thus, long-term continuity in their creation. (Length of large anthropomorph at top right: about 1.5 m. Photo by D.S. Whitley).

Glacial erratics on the Northern Plains (Keyser and Klassen 2001) and the Bonneville flood debris on the Columbia Plateau (Pavesic 2007) have both been identified as common locations for rock writing sites. This association probably results from the fact that these landscape features are otherwise out of place relative to their larger geological contexts, marking them as anomalous. These two examples may parallel the Native Californian and Great Basin belief concerning ostensibly natural objects which appear to have agency and a will to act and, thereby, are considered supernaturally potent. Rocks that move downhill are a common example of this phenomenon (Bean 1975, p. 26) and they were, accordingly, appropriate places for vision questing and rock writing. Basalt talus or scree slopes are the most characteristic geomorphological features that illustrate this principal, especially in

the volcanic landscapes of eastern California and the Great Basin, thereby combining the inherent power of volcanic features with the assumed agency of moving boulders. It is perhaps then not surprising that these landforms were sometimes used for burials (Lowie 1924; Kelly 1932; Steward 1941; Stewart 1941; Whiting 1950; Fowler and Fowler 1971), including shamans' burials specifically, which were "bringers of power to others in obscure ways" (Shimkin n.d.). Some talus slopes also have vision quest pits (Carroll 2007) and they often have rock writing (Figure 4).

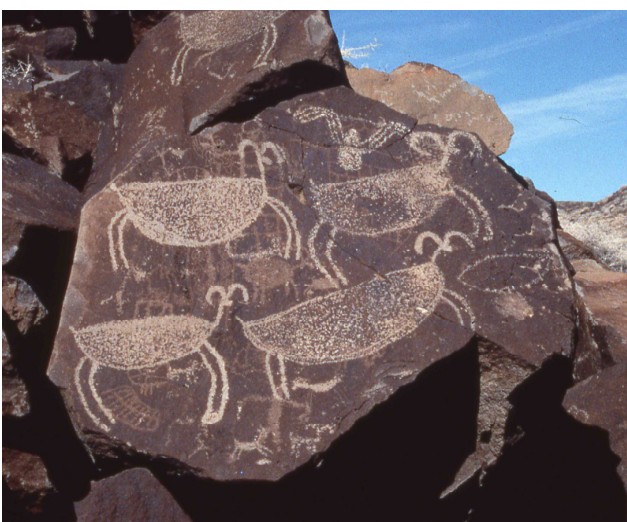

**Figure 4.** Talus slopes comprising rocks that move and, thus, exhibit agency are common locations for rock writing in the California deserts and Great Basin, especially in basalt landscapes. This example, from Big Petroglyph Canyon, Coso Range, shows four bighorn sheep superimposed over earlier/older motifs; bighorns are the most common species portrayed in this locality. They were spirit helpers for rain shamans. (Length of lower right bighorn: approximately one meter across. Photo by D.S. Whitley).

*3.2. Hydrological Features*

According to Miller, water "is the keystone of [Great] Basin religion because power, with its affinity for life, was strongly attracted to" it (Miller 1983a, p. 232). Water more generally was, of course, lifegiving in the physiological sense, but it also had a strong symbolic significance and ontological meaning. It was understood as "a purifying agent . . . spoken of as being like the human breath" (Whiting 1950, p. 40) and it had curative powers (Hudson and Underhay 1978). Shamans bathed in water prior to their vision quests (Kelly 1932, 1936; Driver 1937; Park 1938; Harris 1940; Steward 1941; Aginsky 1943; Gayton 1948; Whiting 1950; Zigmond 1977, 1980; Hultkrantz 1987), in part because spirits lived in water (Bean 1975; Blackburn 1975):

> Springs, and water in general, take on symbolic as well as life sustaining functions. Water itself is a sacred substance to Southern Paiute people, and it must always be approached as a living thing, which means prayerfully. It has its own spirit, and there may also be other specific spirits that live in springs and other water sources that need to be carefully considered . . . Old People say that [spirits], like people, had preferred homes—certain springs that they preferred and where they stayed. People knew where these were and always approached these very cautiously and with utmost respect. (Fowler 2002, pp. 7–8)

One such spirit was Water Baby. This was a dangerous female being believed able to travel through underground rivers from water source to water source. Water Baby was an important Numic shaman's spirit helper, making springs, lakes and other hydrological features appropriate places for vision questing: "Sleeping at certain places where there is water gave dreams of a water baby" (Steward 1941, p. 258; see also Kelly 1932; Steward 1943).

The bubbling of the springs was described as the Water Baby's breath, while hot springs were caused by their cooking fires (Park 1938; Fowler and Fowler 1971; Miller 1983a; Van Vlack 2018).

Like rocks, water sources were in fact portals into the supernatural realm (Blackburn 1975; Whitley 1994b, 2000) with the visionary experience required to enter this world often described as passing into or out of water. As Applegate (1978, p. 118) noted, "water is a powerful supernatural medium which can transform persons and objects from one ritual status to another." One prospective shaman's visit to the supernatural was described as follows:

> At dusk [he] met two strangers, who took him with them into the stream, through two doors, one formed of a snake, one of a turtle. He had become unconscious. Inside their house the otters, for such they had become, resumed human shape. They offered power to their guest, with the threat that he could not live if he refused. He took the gift, but asked for instructions concerning it. (Kroeber 1925, p. 514)

The conceptual equivalence of an entry into a rock versus into water with both as portals to the supernatural is established in the following Numic account of a shaman's visionary experience, where the two occur at one location:

> [A man] ate tobacco and got drunk on it [i.e., hallucinated]. At the hole there was a rock that opened and closed. He waited, and at a moment when the rock opened, he slipped through quickly and went in. . .The man saw water that was like a window. He could see the mountains through it. But it wasn't water. He passed through it and did not get wet. When he was outside, he looked back and saw the 'water' again. (Zigmond 1980, p. 177; quote in original)

Rock writing sites, accordingly, are often associated with springs and other bodies of water (Figure 5) while the association between water and the supernatural realm is, in other cases, signaled by the use of so-called aquatic motifs on the panels (e.g., Hudson and Conti 1981; Whitley 1992; Figure 6).

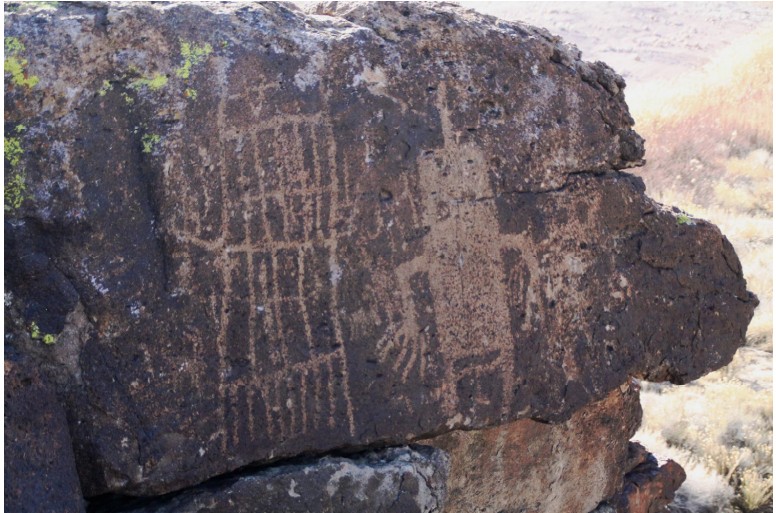

**Figure 5.** The Pahranagat (Southern Paiute for "feet sticking in water") Valley is an unusually well-watered region in the otherwise very arid Mojave Desert of southern Nevada. It is home to the Pahranagat National Wildlife Refuge, a key location on the Pacific Flyway for migratory birds due to its substantial lakes and wetlands. This panel of petroglyphs is from Black Canyon, within the refuge. It shows a common motif pairing in this area: a patterned-body anthropomorph, or shaman figure, on the left, and an engraving identified by informants as a Water Baby on the right—a powerful female spirit helper who lived in water sources and could be obtained as a spirit helper by vision questing at such locations. (Height of Water Baby figure: approximately 0.75 m. Photo by D.S. Whitley).

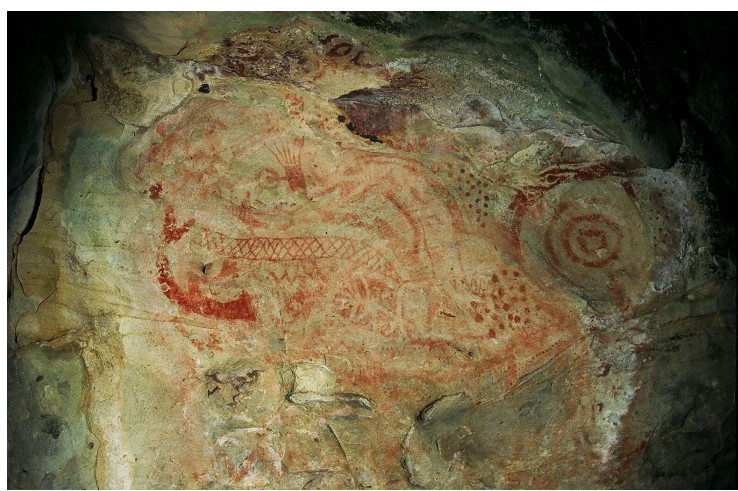

**Figure 6.** The association between the supernatural world and water is also signaled by aquatic motifs. This panel, from the Chumash site of Arrowhead Spring, is immediately above a small seep and pool. It shows a hammerhead shark–anthropomorph (dark red figure on far left) and salamander (central figure, with head at the bottom). (Length of salamander: approximately 0.75 m. Photo by D.S. Whitley).

### 3.3. Transient Natural Phenomena

Landscape features alone were not the exclusive indicators of supernatural power. Certain unusual events, including ephemeral geophysical and meteorological phenomena, were also considered expressions of potency (Waterman 1910; Bean 1975; Miller 1983a). Repeated, even if sporadic, occurrences at a particular locale could cause that spot to be associated with the respective power, making the location itself sacred. The most important transient phenomena were earthquakes and luminous aerial displays.

Earthquakes are, of course, famously common in California. But tremors were understood as having supernatural causes and implications across the west coast, beyond California alone (e.g., Carver 1998; McMillan and Hutchinson 2002; Ludwin 2001; Ludwin et al. 2005; Ludwin and Smits 2007; Thrush and Ludwin 2007; Hough 2007; Ruuska 2014; Stoffle 2022). In northwestern California, in fact, earthquakes were understood to result from a cosmic imbalance. The Yurok mid-winter World Renewal Ceremony was conducted in part to restore equilibrium and prevent this natural disaster from occurring (Kroeber 1907, 1976).

Tribal groups throughout south–central and southern California believed that seismic events were caused by the actions of powerful subterranean spirits (Hooper 1920; Spier 1923, 1930; Drucker 1937; Driver 1939; Reid 1958), conceptualized as the movements of two large snakes by the Chumash (Blackburn 1975). Shamans were thought able to predict, control and sometimes even cause earthquakes due to their relationships with these supernatural beings and powers (DuBois 1908; Kroeber 1908b; Gayton 1948; Heizer 1955; Blackburn 1975; Zigmond 1977). Places experiencing frequent tremors or earthquake swarms, such as rift zones and major faults, were themselves then associated with these spirits and their powers.

The ritual importance of earthquake rift zones in Native California is well demonstrated by the renowned San Andreas Fault, the longest such seismic feature in the world. It cuts southeast—northwest through the Transverse Ranges, crossing immediately north of Mount Piños, the center of the Chumash world and their creation point. It then trends north/northwest through the Carrizo Plain, ultimately diving into the Pacific Ocean along the Mendocino coast, covering a total of about 800+-miles (~1300-km) along its path. The Chumash called the rift zone in the vicinity of Mount Piños the *'Antap* plain. *'Antap* was a word otherwise signifying ritual official. But it was also used to refer to a ritual corridor from which dancers would emerge during ceremonies (Applegate 1974, p. 199), with these

dancers said to come, 'mysteriously' (meaning due to shamanistic powers) from Mount Piños itself (cf. Hudson et al. 1977, p. 43). Given the association of this mountain with the Chumash origin, this suggests that they came directly from the mythic creation for their ritual appearances.

More directly, the *'Antap* plain area was:

> one of the last strongholds of native sorcery [i.e., shamanism]. It was said that spirits lit their fires and began to dance there at dusk. (Applegate 1974, p. 199)

> You hear bullroarers and *tocar* [instruments playing] and *gritar* [shouting]—dogs barking—many people in there—it is like a fiesta. . .The wind blows strong, the earth quakes. (Hudson and Underhay 1978, p. 41; cf. Blackburn 1975, p. 299)

Bullroarers, flutes and rattles, importantly, were used to demarcate the beginning of rituals and, thus, entry into sacred time (Applegate 1974, p. 199; Hudson and Blackburn 1986, p. 318). Hearing them implies that a supernatural event is occurring. Strong winds, known as "the breath of the world," marked the appearance of spirits including especially Thunder, for "thunder was the wind" (Blackburn 1975, p. 96). Thunder in fact brought spirit helpers to shamans (Blackburn 1975, p. 273).

It is important then to note that earthquakes themselves are more than just seismic incidents. They also may have pronounced sonic effects (e.g., Slemmons 1957), as I can personally attest: I was awake for the 1994 Northridge 6.7-moment-magnitude quake which arrived where I lived at 4:30 a.m. Although I did not immediately understand what was about to happen, I heard the quake coming. (It sounded like a sudden, massive windstorm, almost like a train, coming my way before the shock wave hit; the house rocked off its foundation, the books fell from their shelves, and every glass, bottle and piece of ceramic hit the floor and shattered). Reference to the wind blowing, loud noise and the earth shaking in Chumash descriptions of the *'Antap* plain then emphasizes, on multiple levels, its connection to earthquakes and supernatural potency. Put another way, entry into sacred time, as demarcated by the whirring of the bullroarer, was intended to mimic the start of a temblor and entry into the supernatural.

A second kind of transient natural phenomena signaling supernatural power involves luminous aerial displays: lightning, "earthquake lights," and the less understood "ball lightning." The association of these fleeting events with expressions of power parallels the shamanic correlation between the supernatural and quartz crystals. Quartz generates triboluminescence, a photon glow, when rubbed or broken. This was interpreted as a visual manifestation of power with quartz intentionally broken on vision quests to generate this potency and quartz hammerstones preferentially employed to engrave petroglyphs due to the inherent power of this stone (Whitley 2001; Whitley et al. 1999).

Although lightning can occur anywhere, its distribution overall is patterned, following topography and the regular paths of prevailing seasonal storms. Strike density and frequency in California are greatest in the California deserts; in the mountains between about 2700–3000 m elevation (not necessarily on the highest peaks); and during the summers due to monsoonal storm systems (Van Wagtendonk and Cayan 2008). Basalt flows with magnetite and other magnetic minerals, and basalt scarp edges, may also preferentially "attract" lightning (Cummins 2012; Mayet et al. 2016). And then lightning is also common during volcanic eruptions (McNutt et al. 2010; Cimarelli and Genareau 2021; Van Eaton et al. 2023), further associating it with specific locations, landscape features and events—including ones already considered imbued with power due to their other characteristics.

Lightning and associated thunder were widely linked to supernatural power throughout far-western North America. Thunder brought spirit helpers to shamans for curing rituals, for example, among the Chumash (Blackburn 1975, p. 273), while lightning did the same for the northeastern Californian Klamath (David 2016). An important supernatural spirit for the southern California Takic speakers was Takwitc ("Tahquitz") who manifested as a luminous aerial event. He was the mythic first shaman, thought a cannibal spirit (i.e., a stealer of souls) and was, therefore, dangerous, could award shamanistic powers

and lived inside a rock or cave (Gifford 1918; Hooper 1920; Patencio 1943; White 1963; Bean 1974, 2017). This was believed to be at the eponymous Tahquitz Peak (Bean 1974; Harrington 1978; Bean et al. 1991) in the San Jacinto Mountains, above Idyllwild.

Tahquitz Peak, at 2962 m, notably falls within the elevational band for the greatest density and frequency of lightning strikes (as effectively does Mount Piños at 2697 m). The nature of the luminous event signaling the presence of Takwitc, however, is uncertain, with descriptions including a meteoric fireball (DuBois 1904, 1906); a 'bright' or 'beaming' light flying near the surface of the ground (Waterman 1910, p. 79; Kroeber 1908a, p. 65); a meteor or shooting star (Sparkman 1908); an electric fireball (Gifford 1918); a low flying meteor (Benedict 1924); and "ball lightning" (Waterman 1910). Most subsequent researchers have accepted Waterman's interpretation of it as ball lightning (e.g., Harrington 1942; Van Valkenburgh 1952; Blackburn 1975; Beemer 1980; True and Waugh 1986; Whitley 2000), but the nature of ball lightning itself is uncertain, including even its existence (Holle and Zhang 2023). Most likely is the possibility that the sighting of any type of transient aerial display was thought a potential manifestation of Takwitc specifically and supernatural power more generally.

So far unmentioned in this regard is the possibility that the luminous events might also have resulted from earthquake lights, or EQL: luminous displays that may occur before and during, but not after, a major temblor (Derr 1973; Lockner et al. 1983; Fidani 2010; Witze 2014). Ludwin (2001), for example, identified three indigenous northwest tales that associate lightning with megathrust earthquake events, one of which is from the northwestern California Yurok. She interprets each description as more correctly pertaining to EQL.

EQL is thought caused when electrical charge conductors in certain metamorphic and igneous rocks are subjected to stress, turning them into semiconductors (Thériault et al. 2014). They last from a few seconds to several minutes, with pre-shake manifestations typically stationary or moving globular displays (like descriptions of ball lightning), whereas those during the quakes commonly appear as short flashes of flame-like light. (cellphone photos and videos of EQL may be found on the internet). EQL associated with the famous 1811–1812 New Madrid, Missouri, earthquake sequence were visible from as far as 600 km away while those generated by the 1906 San Francisco earthquake were seen from a distance of about 100 km (ibid).

Earthquakes then generate not only earth movement but potentially also unusually loud noises and aerial light displays, any one of which was a sign of supernatural power. This concatenation of phenomena, like the association of lightning storms with volcanic eruptions, almost certainly heightened the perceived potency of the locations where they occurred.

The San Andreas rift zone, notably then, contains the two largest concentrations of Chumash rock writing (Whitley et al. 2007; Whitley and Whitley 2012): the San Emigdio Ranch (named after Saint Emygdius, the patron of earthquakes), a short distance north of Mount Piños; and the Carrizo Plain, further to the northwest. The Carrizo is the classic location for geological studies of the San Andreas rift zone due to the well-exposed fault scarp and offset drainages that form its eastern boundary (e.g., Zielke et al. 2010). These two locations are renowned for their spectacular examples of Chumash rock paintings (Figures 1 and 7). As observed a century ago: "The most remarkable pictographs [in California] are those in Chumash country, beginning with the famous Corral [Painted] Rock in the Carriso [sic] Plains, the largest and most notable group in the state" (Kroeber 1925, pp. 937–38).

It is difficult to directly link the locations of rock writing sites with aerial light events given the fleeting nature of these phenomena. But petroglyphs are sometimes present on lightning-struck boulders (Benson and Buckskin 1991; M. F. Walker 2007; Shimkin 1986), suggesting that the locations of these strikes were recognized as appropriate supernaturally powerful places for rock writing rituals.

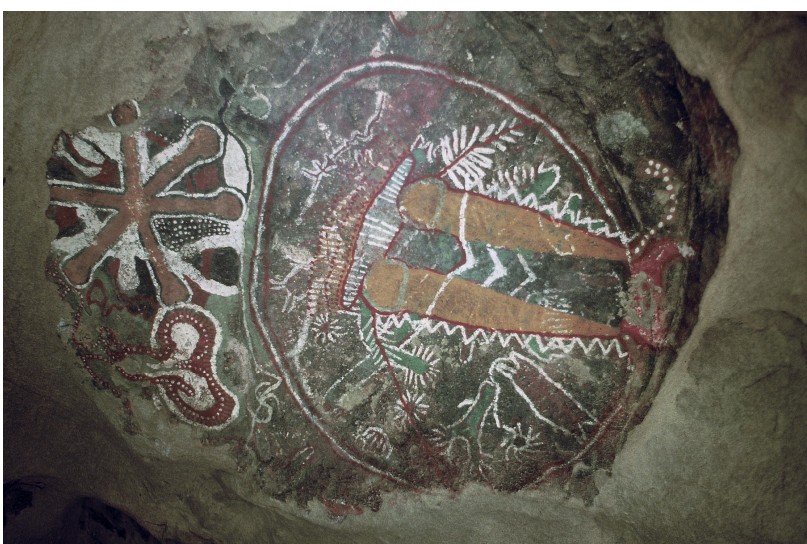

**Figure 7.** The rock writings on the San Emigdio Ranch and the Carrizo Plain are both located in the San Andreas Fault rift zone and they represent the largest concentration of pictographs in Chumash territory, if not in Native California as a whole. The Pleito Creek Cave is a small but especially spectacular example. Kroeber (1925) was the first to note the psychedelic quality of these paintings, suggesting that they may be a product of hallucinatory experiences (cf. Blackburn 1977). (Diameter of cartouche on right: approximately 0.5 m. Photo by D.S. Whitley).

*3.4. Cultural Associations*

Cultural factors too played a role in the determination of ritual locations, including rock writings, although, again, these instances were also understood in terms of ontological beliefs about the distribution of power. The most important of these variables involved the fact that power was attracted to life and was, therefore, necessarily concentrated where lifeforms congregate (Bean 1975; Miller 1983a). Villages, where numerous rituals were conducted, are the primary example here. Villages were also located at or near water sources, the abode of spirits, further emphasizing their appropriateness for rock writing ceremonies. Many examples of rock writing, accordingly, are found in or adjacent to villages (Figure 8).

A recently published article about Pinwheel Cave (Robinson et al. 2020), one of the cluster of San Emigdio sites associated with habitation deposits, warrants discussion in this regard. Despite a wide consensus about the relationship of Chumash rock writing to shamanism (e.g., Applegate 1975; Blackburn 1977; Garvin 1978; Hudson and Underhay 1978; Hudson and Lee 1984; Hudson and Blackburn 1986), (Robinson 2013; Robinson et al. 2020) has repeatedly questioned the origin of much rock writing in shamanic vision quest-ing based on the presence of the sites in villages. He falsely claims that I have contended that the sites were the exclusive domain of shamans (Robinson 2013, p. 61) and, therefore, neces-sarily removed "from normal activity of the wider population" (Robinson et al. 2020, p. 1). As is instead clear, Robinson fails to understand the anthropological concept of ritual "avoidance," as described specifically for Native Californian rock writing sites in the Zig-mond (1977, p. 79) quote above. (Men were also required to "avoid" their mother in law, including by never talking directly to them, even while they potentially shared a hut). He also selectively ignores the fact that I first highlighted the association of rock writing with villages over three decades ago and have continued to do so, repeatedly (e.g., Whitley 1987, 1992, 2000; Whitley et al. 2007; Whitley and Whitley 2012); that I identified the potential for private ceremonies at villages sites during the dispersal phase of the seasonal round, when villages were unoccupied and when vision quests were often conducted (Whitley et al. 1999, p. 16; cf. Hultkrantz 1987; Van Vlack 2012b); and that I have also described the use of the sites by non-shamans (Whitley 2000; Whitley and Whitley 2012; Whitley 2022),

including a healing ceremony that I participated in during the 1990s. There is nothing contradictory about the use of village sites for a variety of kinds of rituals, including private ceremonies that were the origins for rock writing. Nor does this somehow imply that secular activities were excluded from these same ritual locations, including food preparation and consumption, as Garvin (1978) first pointed out almost a half-century ago.

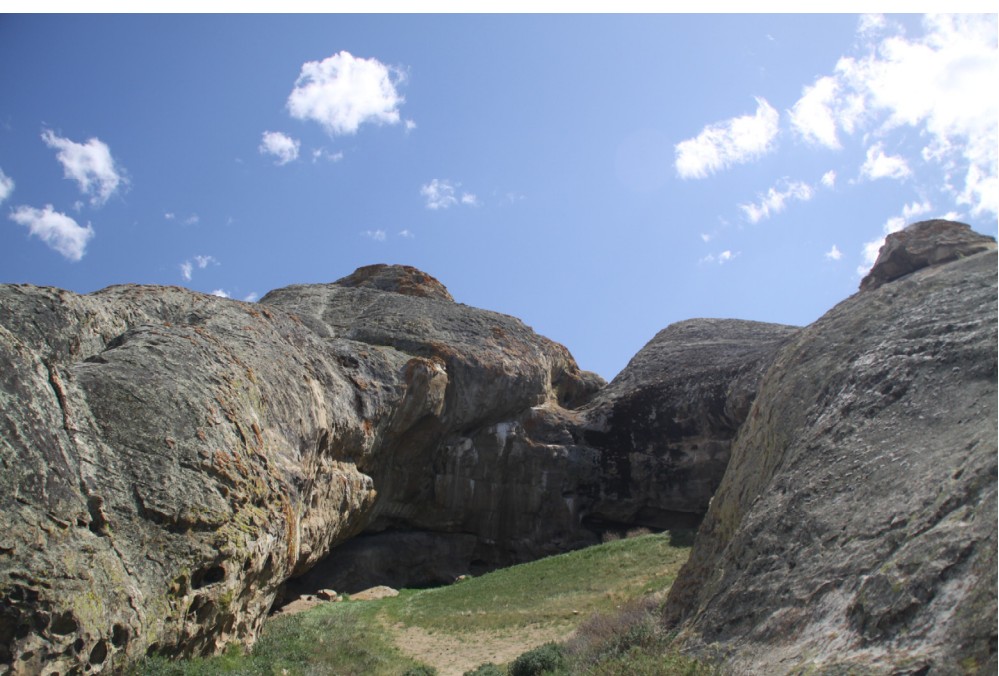

**Figure 8.** Villages, occupied during the aggregation phase of the seasonal round, concentrated power because power itself is attracted to living things. Many such habitation sites, accordingly, have associated rock writing. Seasonal occupation followed by abandonment facilitated the use of these locations for shamanic vision questing. Perhaps the best example of this co-occurrence is Painted Rock, Carrizo Plain, which has a midden deposit within its amphitheater-shaped interior (approximately 27 m wide and 33 m deep) as well as around the outside of this distinctive outcrop. Many of its interior and exterior walls have rock paintings[2]. (Photo by D.S. Whitley).

The presence of jimsonweed (*Datura wrightii*) quids at the Pinwheel site, as documented by Robinson et al. (2020), instead clearly demonstrates a connection between Chumash rock writing and the ingestion of hallucinogens, that is, with shamanistic practices, as Robinson has been forced to finally acknowledge. Paradoxically, he nonetheless continues to reject a potential origin for the pictographs in shamanic vision questing based on his interpretation of the pinwheel-like motif at the site as an opening *Datura* bud. He asserts that a shaman's rock writings are, somehow, necessarily limited to "shamanic self-depiction" and "private images seen in trance" which could not include such a motif (ibid., p. 10)—for reasons he does not explain or otherwise support. Yet as Applegate (1975, p. 12) observed, "The *Datura* drinker might see many other things besides a dream helper." Among the neighboring Yokuts, moreover, the ritual officiants who prepared and served jimsonweed necessarily had *Datura* as their spirit helper (Gayton 1948, p. 38), indicating that shamanic visions of *Datura* were well known in the region.

Robinson et al. then contend that "rock art here, instead, is an active accompaniment to the trance event, communicating key information about the constituent elements of such events within communal contexts" (Robinson et al. 2020, p. 10). That the motif may have been directly associated with an altered state of consciousness experience, and that it may have also transmitted details about such an event to the population at large, have no bearing on whether or not it was created by an individual shaman during a vision quest. Such a contorted strawman claim, in contrast, confuses and conflates the origin

versus meaning of rock writings, their initial or primary versus subsequent secondary uses and understandings, as well as the fundamental role of graphic imagery in social communication. As noted above, Driver in fact recorded, that shamans: "painted their spirits on rocks 'to show themselves, to let people see what they had done'. The spirit must come first in a dream" (Driver 1937, p. 86; quote in original). And as I noted almost two decades ago, moreover, the ethnography suggests that a shaman's rock writing site "may have been used, during male initiations, to give the boys an idea of the appearance of the supernatural that they themselves were about to visit under the influence of *toloache* [*Datura*]" (Whitley 2006, p. 323). Such a secondary use seems likely for the Pinwheel site given the presence of jimsonweed quids.

The archaeological record at Pinwheel Cave provides compelling evidence linking the Chumash rock writings with shamanistic rituals, practices and visionary experiences. That this site is located in a cave with an associated habitation deposit, in the San Andreas rift zone area, demonstrates its appropriateness for pictographs based on the multiple variables that contributed to the perception of this locale as supernaturally powerful.

Two additional cultural factors affected the selection of particular locations for rock writing. The first concerned the fact that power could reveal itself through individual experiences and these effectively could happen anywhere:

> Power remains diffused everywhere while also concentrated in web-like pathways. It can still be encountered accidentally while travelling, provided that close attention is paid to surroundings. (Miller 1983a, p. 82)

In theory, at least, any location could then be a supernatural portal appropriate for rock writing, if the correct experience occurred there. The second and related factor concerns ritual reactions to encountering previously identified sacred places:

> When ceremony or power seeking is successful at such places, they are selectively marked so future human visitors can more fully understand the purpose of the place. (Stoffle et al. 2011b, p. 11; cf. Carroll 2007)

As Carroll et al. then explain: "The logic was simple—*if knowledge resides in powerful places, then let us return to those places where we can recapture it*" (Carroll et al. 2004, p. 141; emphasis in original). Rock writing, as one type of ritual activity, left a permanent record of a supernatural experience. This served to promote future uses of this same place for similar ceremonies and experiences. Rock writing, in this sense, begets more rock writing, as permanent signs of devotional behavior. Concentrations of rock writing motifs are then to be expected, and are common, in the archaeological record.

The features and phenomena identified above represent a significant number of signs of supernatural power but, in fact, not all of the possibilities. Additional potential indications of potency include pigment outcrops, obsidian sources (which, of course, are volcanic landforms) and plants and animals themselves associated with power, including jimsonweed, native tobacco, stinging nettles, rattlesnakes, red harvester ants and horned lizards (cf. Van Vlack 2012a; Van Vlack et al. 2013; Stoffle et al. 2015). But with the exception of pigment localities and obsidian, these are essentially ephemeral and transient and of limited value to archaeological analysis.

## 4. Discussion

Western scientists have traditionally perceived religion and its associated beliefs and rituals as irrational and, thus, the polar opposite of rational Western science (e.g., Dawkins 2006; Dennett 2006). Because science and religion are fundamentally incommensurable, religion to many such researchers is viewed as simply not worthy of scientific study. In a series of important books, moral philosopher Mary Midgley (2001, 2002, 2003) has argued that this attitude developed because Enlightenment science for centuries was in competition with religion as a mode of thought. Although this historical conflict certainly has contributed, I believe that more is involved here than is suggested simply by a debate over the rationality of science versus the putative irrationality of religion. Part of the

debate instead stems from a fallacy of equivocation—a conflation of two distinct issues: the epistemic problem, whether the gods and, therefore, religions are 'real;' versus the separate question about the nature of religious thought. This issue boils down to whether religious beliefs and practices are primarily 'mystical,' as Lucien Lévy-Bruhl (e.g., Lévy-Bruhl 1923; contra Malinowski [1925] 2014) argued and, therefore, beyond the bounds of scientific understanding or, instead, whether they involve systematic, logical and, therefore, rational thinking. If we allow the latter, we have a potential avenue for understanding religions, or at least aspects of religions, scientifically. This has important implications for a discipline like archaeology which has struggled to interpret pre-contact religions despite the fact that the evidence for rituals is a common component of the archaeological record.

As I have illustrated above, religious beliefs and practices may in fact involve systematic, logical thought, regardless of the ultimate truth value (and rationality) of the underlying theological percepts (see also Whitley 1994b, 2000, 2008, 2021, 2024). Native Californian and Great Basin ontological beliefs about supernatural power and its distribution on the landscape are one example of such a rational cognitive system. Once deciphered, this affords us an opportunity to understand the logic of the distribution of ceremonial practices on the land, not only explaining why religious features are found in certain places and not others but also predicting where additional examples may be found in the future. Rock writing sites provide my example. These are typically found in certain contexts: specific types of geological formations and geomorphological landforms, including caves, rockshelters, volcanic formations and talus slopes, but they are rarely found on mountain peaks. The sites are also commonly found near water sources such as springs and at villages which, due to the presence of people, necessarily also concentrated supernatural power.

Ontological beliefs about material landscape features alone do not fully explain the distribution of ritual activities, however. Transient or ephemeral phenomena were also perceived as expressions of power and the places where these occurred or were sighted were themselves associated with this power. Earthquakes and aerial light displays (lightning, earthquake lights and ball lightning) are the most important here, and these also figured in beliefs about potency, with rock writing sites clustered in areas that frequently experienced these types of events. These correlations are not perfect of course and some variability is always to be expected. Given that individual revelations could happen anywhere and that such experiences would signal the potency of a specific place, these spots would accordingly also be marked as power locations by rock writing or other ritual features, promoting continued subsequent use.

Perhaps more importantly, the logical ontological thinking involved in understanding the distribution of power on the landscape also explains the major concentrations of rock writing sites, ones with an order of magnitude or more petroglyph and/or pictograph sites and motifs greater than the remainder. The Carrizo Plain, in Chumash territory, provides one example, with approximately two dozen rock writing sites in a relatively restricted area. The key characteristics of this location linking it to power start with the San Andreas fault which runs through this inland valley and the earthquakes as well as earthquake lights and sonic effects that often accompany major tremors. Saline Soda Lake, which filled the valley bottom during pre-contact times, likely also contributed to the perception of this location as unusually imbued with power. Pigment sources are a third sign of supernatural power. But Painted Rock, an unusual outcrop that represents a kind of open-air cave, itself was likely a conduit for supernatural potency (Figures 8 and 9). Based on the archaeological record, so too would have been the large number of villages sites, and thus people who concentrated power, on the plain. The combination of these factors, rather than any single feature alone, helps us understand why the Carrizo has the largest concentration of Chumash rock writing.

An even more impressive example—in the sense of the numbers of sites, their size and the total quantities of motifs—is the Coso Range, eastern California. Credible estimates suggest that there may be a million or more petroglyphs in this locality, dwarfing all other rock writing concentrations in North America and rivaling, if not exceeding, the

other largest localities in the world. The supernaturally powerful features of this location, among others, include a dramatic volcanic landscape with 38 rhyolite domes and cinder cones, talus slopes and basalt flows; large hot springs with multi-colored, sometimes boiling mud pots and pools; fumaroles and associated pigment sources; a small lake; and active seismicity due to its situation on the Walker Lane–Eastern Shear Zone fault complex (Figure 10). Coso Hot Springs, in fact, was "the most powerful of many healing springs in the west" (Fowler et al. 1995, p. 49), from which all others were derived (Stoffle et al. 2011a). Put another way, the Cosos were the most sacred locality in the far west, due to the synergistic relationship between these individually potent landscape features and characteristics. This explains why it was used for rituals by shamans from as far away as Fort Duquesne, Utah, by the Yokuts from the San Joaquin Valley and the Tubatulabal from the Sierra Nevada as well as by local Shoshone and Northern and Southern Paiute peoples.

Although all aspects of pre-contact religions are not likely archaeologically knowable, I maintain that many facets may be, more than most archaeologists previously might have allowed. Understanding these characteristics of past ritual and belief will start when we finally move beyond "the assumed superiority of the present over the past, and the rule of the European possessors of reason over the 'primitive' inhabitants of the 'undeveloped' world" (Hawkes 2003, p. 136; quotes in original). It is then necessary to emphasize, as Claude Lévi-Strauss (1966, p. 23) reminded us, that "man has always been thinking equally well." Native Americans have always *thought like* Western scientists, in terms of logic and reason—our cognitive systems are the same—even if we have different knowledge bases and seek explanations in different ways. We will come to a much better understanding of pre-contact Native American lives, including their religion, if we use this certainty as our starting point for investigations.

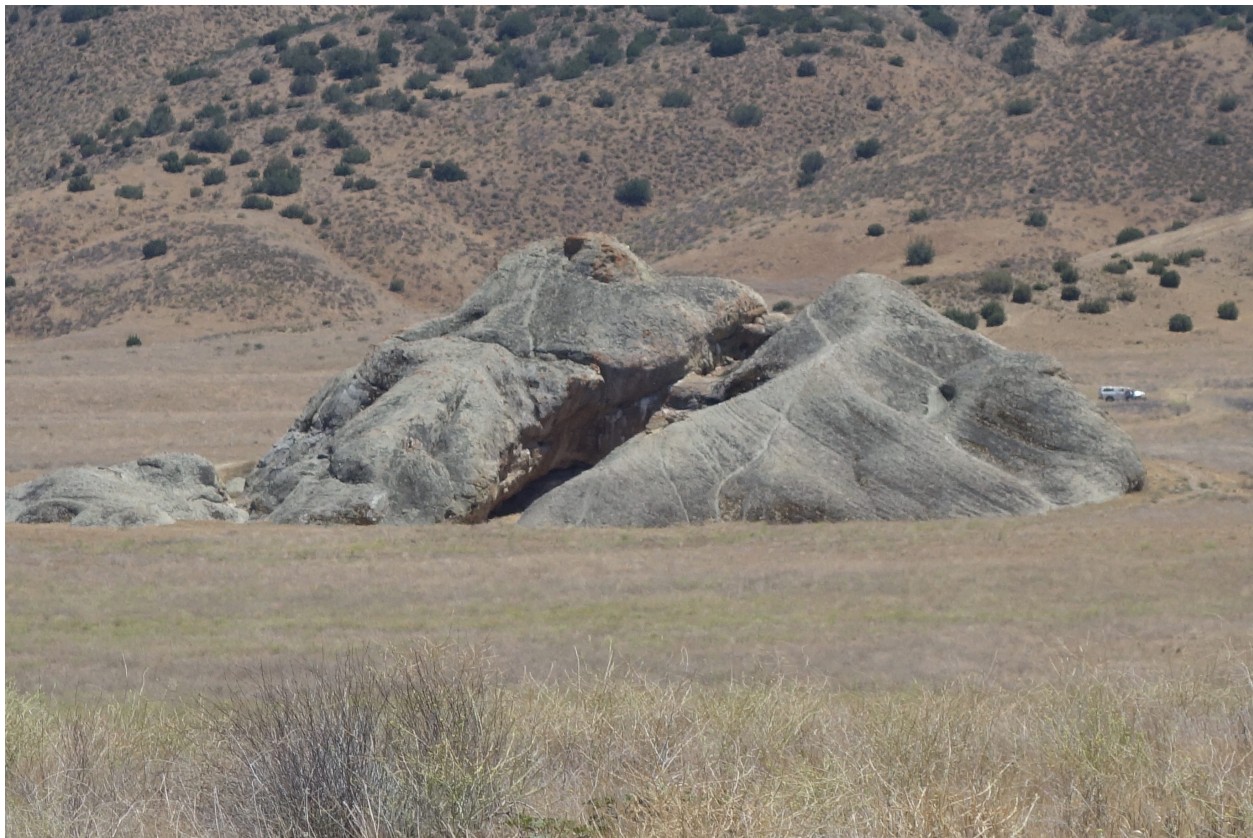

**Figure 9.** Painted Rock, Carrizo Plain (see also Figure 8). This isolated outcrop has an unusual interior amphitheater—an open-air cave—and outside cliff faces (rear side of this photo) with numerous pictographs. A tinaja at the top holds water and further links this feature to supernatural power. (Photo by D.S. Whitley).

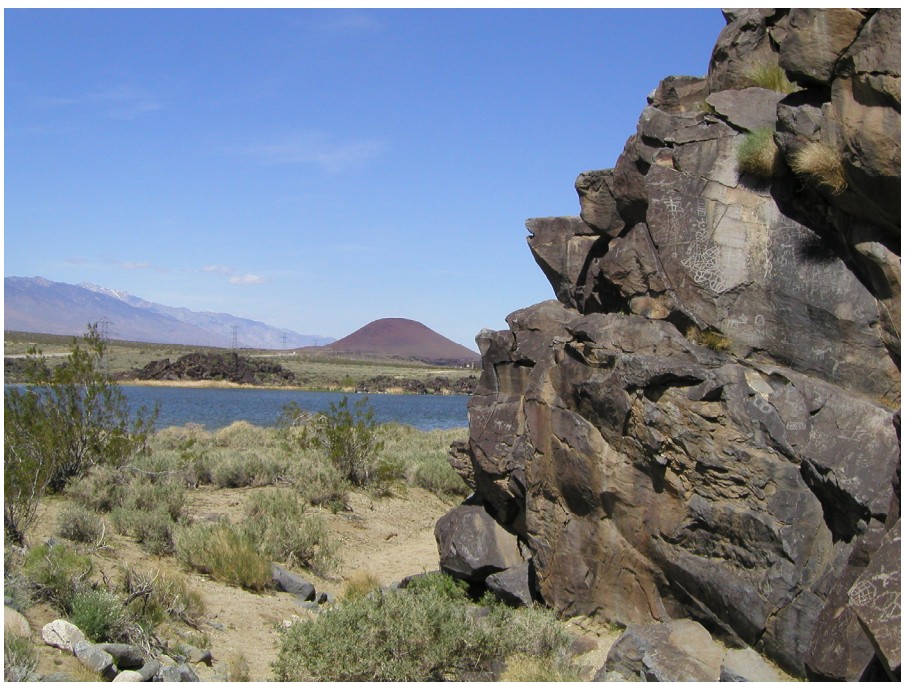

**Figure 10.** No place in the far west was more powerful than the Cosos due to its combination of supernatural features and phenomena, a circumstance resulting in its massive number of rock writings. The natural lake shown here, Little Lake, is on the western edge of the Cosos. A winter village and a series of petroglyph localities are located along the lake margin. Red Hill, a volcanic cinder cone, is in the background. (Photo by D.S. Whitley).

**Funding:** This research received no external funding.

**Data Availability Statement:** Data relevant to this study is available via the sources cited above.

**Conflicts of Interest:** The author declares no conflict of interest.

## Notes

<sup>1</sup>    I use the term 'tribes' following contemporary Native American, rather than formal anthropological, usage. A tribe is, thus, a generic term for an ethnic (and sometimes linguistic) group that resides in a particular region and may have a variety of kinds of organizational structures. I also use 'rock writing' as opposed to the more common 'rock art' because rock writing is the much-preferred label for petroglyphs and pictographs among the tribes with whom I work.

[1]    I use the term 'tribes' following contemporary Native American, rather than formal anthropological, usage. A tribe is, thus, a generic term for an ethnic (and sometimes linguistic) group that resides in a particular region and may have a variety of kinds of organizational structures. I also use 'rock writing' as opposed to the more common 'rock art' because rock writing is the much-preferred label for petroglyphs and pictographs among the tribes with whom I work.

[2]    Due to the rampant exploitation and commodification of images of the Carrizo Plain pictographs, the Bureau of Land Management has requested that no additional photographs of the rock writing motifs and panels in the Carrizo Plain National Historic Landmark Rock Art District be published. No images of the paintings from the district are included here as a result.

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
