# Peer review of "Ontological Beliefs and Hunter–Gatherer Ritual Landscapes: Native Californian Examples"

_religions, doi:10.3390/rel15010123_

Round 1
Reviewer 1 Report
Comments and Suggestions for Authors
Although not necessary to add to the current paper, it is worth noting that water is mediatory both in terms as a physical medium and as a metaphorical construct. Water is essential to physically remove rock when creating petroglyphs and to allow pigment soaking into the rock when creating pictographs. When damp, most petroglyphs and pictographs become more visible. A careful reading of ethnographic sources will reveal that areas associated (or believed to be associated) with water and dampness, visible on the ground or hidden below ground (or even absent but believed to present in a hidden and reversed spirit world) are the places where spirits can be contacted. Medicine people tend to watery places before approaching spirits, both to purify but also because entering the world of spirit beings is often expressed as entering a damp or watery realm.
Author Response
Thanks you for these comments.
Reviewer 2 Report
Comments and Suggestions for Authors
In my opinion this is an important and thought-provoking paper. It covers a great deal of literature, some of it not available or not known to many scholars, and thus it shed new light on the rationale behind the selection of rock writing (I like the term!) locations and the fact that in some cases huge concentrations of depictions are to be found at specific locales, but not in others. The fingerprints of David Whitley are easily recognized in the blind-review manuscript, and I believe I am not completely wrong assuming he is the author. His long lasting research on rock depiction and shamanism is to be admired, and I am glad I got a chance to review this interesting paper that presents a new and innovative phase of his contribution to the field.
Seeing the belief systems of indigenous groups as logical, following an ontological and cosmological conception of the world, is an important step forward in understanding the archaeological heritage left by our ancestors, and this paper makes a valuable contribution in this direction. I highly recommend it for publication in the journal Religions.
Below are some general and specific comments, to be brought to the attention of the author. I do not suggest these as a must, but leave it to the author to decide to take it or not. I think the paper is very good the way it is, and my intention 1s only in providing some food for thought:
Line 29: Indeed little effort in this regard was made in the US, but this is not the case in Europe and Africa. I would add some of these for a broader perspective. I know the author is very well familiar with this kind of literature and the reader should be advised accordingly, I believe.
Line 36-41. Indeed this was the approach in the US, but not so much amongst some European colleagues, that presented appreciation towards our earliest ancestors, for example Henri Breuil and others. I think again that the picture better be a bit balanced.
Line 121-2: I believe it is now rather widely accepted that HG modified and altered the land since time immemorial. It is also accepted that animals also modify the landscape, in a concept well known as niche construction. I advise rephrasing this statement.
Line 146: Is the supernatural power some sort of equivalence to the concept of “Animal Master”? There is a lot of literature on this phenomenon, and maybe mentioning it will better universalize this line of thinking. See for example this recent book: Chacon, R. J. (Ed.). (2023). The History and Environmental Impacts of Hunting Deities: Supernatural Gamekeepers and Animal Masters. Springer Nature.
Line 169: I would remove “nightmares”. This is in the eye of the beholder and seems like a western concept to me.
Line 181: Likewise with “if not terror”. Again, a concept that must be alien to HG.
Line 307: The term “gender symbolism” is not explained and is a bit hard to understand. Some clarifications are in order.
Line 583: Explain EQL
Line 616-618: References are needed for the support of the psychedelic quality of Pleito images by subsequent research.
Lines 643-662: I am glad the debate with Robinson is put forward. I strongly take sides with Whitley for the use of shamanism and an explanatory concept and believe his arguments are strong and convincing. Notwithstanding, I think Pinwheel cave should be specifically mentioned and referred to in the text. It is a very interesting site, and in my view it is a wonderful demonstration of ritualistic use of Datura as mind altering substance for expanding human consciousness, in the framework of shamanistic practices carried out in the cave alongside other more mundane activities. I know this is not what Robinson makes out f that, but the site fits well with the agenda presented in this paper and should be part of it, I believe.
Author Response
Thank you for the thoughtful comments. They are very helpful. I’ve added comments/explanations below:
Line 29: Indeed little effort in this regard was made in the US, but this is not the case in Europe and Africa. I would add some of these for a broader perspective. I know the author is very well familiar with this kind of literature and the reader should be advised accordingly, I believe.
- Qualifiers added to indicate that the discussion is specific to North America
Line 36-41. Indeed this was the approach in the US, but not so much amongst some European colleagues, that presented appreciation towards our earliest ancestors, for example Henri Breuil and others. I think again that the picture better be a bit balanced.
- This paragraph is specific to Great Basin ethnographers, first, and then behavioral ecology which is, by far, the emphasis of current Great Basin archaeological research.
Line 121-2: I believe it is now rather widely accepted that HG modified and altered the land since time immemorial. It is also accepted that animals also modify the landscape, in a concept well known as niche construction. I advise rephrasing this statement.
- This circumstance was acknowledged in the text by the modifiers “traditionally” and “(supposedly)” signaling my rejection of the older described beliefs.
Line 146: Is the supernatural power some sort of equivalence to the concept of “Animal Master”? There is a lot of literature on this phenomenon, and maybe mentioning it will better universalize this line of thinking. See for example this recent book: Chacon, R. J. (Ed.). (2023). The History and Environmental Impacts of Hunting Deities: Supernatural Gamekeepers and Animal Masters. Springer Nature
- The Master of the Game (“Ya’ahwera” in Numic; "Mets Hot" for the Yokuts) is one of the numerous shaman’s spirit helpers in the far west, and thus a spirit who was sometimes visited while in the supernatural. The concept was then present but not as formally or ritually developed as, in comparison, the case for the Tucano in South America, or nearly as important as the other spirit helpers in the far west. I first discussed Ya’ahwera in a 1994 paper but I have only found one example directly connecting Ya’ahwera to a rock writing site. Garfinkel, following my 2000 publication of this connection, subsequently has published a few papers on the site, including in the above book (which I have), where he attempts to generalize it beyond the lone specific case. Had that been warranted, I would have done so over two decades ago. Including a discussion of the Master of the Game here would thus narrow rather than universalize my analysis.
Line 169: I would remove “nightmares”. This is in the eye of the beholder and seems like a western concept to me.
Line 181: Likewise with “if not terror”. Again, a concept that must be alien to HG.
- These descriptors are important because of the near-universal Western perception that the shaman’s visionary experiences were ecstatic when our recorded ethnographic accounts clearly indicate exactly the opposite. Native Americans discussed their adverse emotional effects explicitly and viewed these experiences as distinct from the more common REM dreams. Carobeth Laird, e.g., said that her husband George’s power dreams caused him “intense terror” and they “left him sweating and shaking,” causing him to reject his shamanic calling. Isabell Kelly described them as “dangerous.” To leave out these terms would put a Western spin on the shamanistic state of consciousness—which is prevalent in the vast majority of the literature and which I am trying to correct. (I have discussed what I have called this ‘myth of ecstasy,’ promoted first by the title of Eliade’s book and next by the self-realization/counter-cultural movement of the 1960s, in detail in my 2009 book on the Paleolithic). I thus include them exactly to avoid a Western, presentist interpretation of these events.
Line 307: The term “gender symbolism” is not explained and is a bit hard to understand. Some clarifications are in order.
- Explanation added:
“This apparently results from the general gender symbolism of the landscape versus that of the petroglyph and pictograph sites specifically: rock writing sites as female-gendered places versus mountaintops as male-gendered (Whitley 1994b, 1998), thereby illustrating the common regional use of symbolic inversion to emphasize the transition into the sacred (Blackburn 1975; Applegate 1978); in this case male shamans using a female-gendered place for vision quests.”
Line 583: Explain EQL
- Provided on line 585ff.
Line 616-618: References are needed for the support of the psychedelic quality of Pleito images by subsequent research.
- I’ve removed this statement from the figure caption because it would require more detail than would be appropriate in a caption, though I’ve added a citation to Tom Blackburn’s 1977 paper that supports this conclusion. I have added some of this detail in the discussion of Pinwheel Cave as discussed below.
Lines 643-662: I am glad the debate with Robinson is put forward. I strongly take sides with Whitley for the use of shamanism and an explanatory concept and believe his arguments are strong and convincing. Notwithstanding, I think Pinwheel cave should be specifically mentioned and referred to in the text. It is a very interesting site, and in my view it is a wonderful demonstration of ritualistic use of Datura as mind altering substance for expanding human consciousness, in the framework of shamanistic practices carried out in the cave alongside other more mundane activities. I know this is not what Robinson makes out f that, but the site fits well with the agenda presented in this paper and should be part of it, I believe.
- I’m grateful for this comment. I hesitated to respond in detail to Robinson’s (frequent) criticisms, based on what can only be understood as intentional misrepresentations of my publications, but I concur that a rejoinder is appropriate here—especially given the very high profile of his 2021 Pinwheel Cave paper. I have reworked this discussion adding additional details accordingly, expanding on the significance of Pinwheel Cave, as follows:
A recently published article about Pinwheel Cave (Robinson et al. 2021), one of the cluster of San Emigdio sites associated with habitation deposits, warrants discussion in this regard. Despite a wide consensus about the relationship of Chumash rock writing to shamanism (e.g., Applegate 1975; Blackburn 1977; Garvin 1978; Hudson and Underhay 1978; Hudson and Lee 1984; Hudson and Blackburn 1986), Robinson (2013; Robinson et al. 2021) has repeatedly questioned the ethnographically-documented origin of much rock writing in shamanic vision questing based on the presence of the sites within villages. He falsely claims that I have contended that the sites were the exclusive domain of shamans (Robinson 2013:61) and therefore necessarily removed “from normal activity of the wider population” (Robinson et al. 2021:1). As is instead clear, Robinson fails to understand the anthropological concept of ritual “avoidance,” as described specifically for Native Californian rock writing sites in the Zigmond (1977:79) quote above. (Men were also required to “avoid” their mother-in law, including never talking directly to them, even while they potentially shared a hut). He also selectively ignores the fact that I first highlighted the association of rock writing with villages over three decades ago, and have continued to do so, repeatedly (e.g., Whitley 1987, 1992, 2000; Whitley et al. 2007; Whitley and Whitley 2012); that I identified the potential for private ceremonies at villages sites during the dispersal phase of the seasonal round, when villages were unoccupied and when vision quests were often conducted (Whitley et al. 1999:16; cf. Hultkrantz 1987; Van Vlack 2012a); and that I have also described the use of the sites by non-shamans (Whitley 2000; Whitley and Whitley 2012), including a healing ceremony that I participated in during the 1990s. There is then nothing contradictory about the use of village sites for a variety of kinds of rituals, including private ceremonies that were the origins for rock writing. Nor does this somehow imply that secular activities were excluded from these same ritual locations, including food preparation and consumption, as Garvin (1978) first pointed out almost a half-century ago.
The presence of jimsonweed (Datura wrightii) quids at this site, as documented by Robinson et al. (2021), instead clearly demonstrates a connection between Chumash rock writing and the ingestion of hallucinogens; that is, with shamanistic practices, as these authors have been forced to finally acknowledge. Paradoxically, they nonetheless continue to reject a potential origin for the pictographs in shamanic vision questing based on their interpretation of a pinwheel motif at the site as an opening Datura bud. They argue that a shaman’s rock writings are, somehow, necessarily limited to “shamanic self-depiction” and “private images seen in trance” which could not include such an image (ibid.:10)—for reasons they do not explain or otherwise support. Yet as Applegate (1975:12) observed, “The Datura drinker might see many other things besides a dream helper.” Among the neighboring Yokuts, moreover, the ritual officiants who prepared and served jimsonweed had to have Datura as their spirit helper (Gayton 1948:38), indicating that shamanic visions of Datura were well known in the region.
Robinson et al. then contend that “rock art here, instead, is an active accompaniment to the trance event, communicating key information about the constituent elements of such events within communal contexts” (2021:10). That the motif may have been directly associated with an altered state of consciousness experience, and that it may have also transmitted details about such an event to the population at large, have no bearing on whether or not it was created by an individual shaman during a vision quest. Such a contorted straw-man claim, in contrast, confuses and conflates the origin versus meaning of rock writings, their initial or primary versus subsequent secondary uses and understandings, as well as the fundamental role of graphic imagery in social communication. As Driver in fact recorded, shamans: “painted their spirits on rocks ‘to show themselves, to let people see what they had done.’ The spirit must come first in a dream” (1937:86; quote in original). And as I have noted previously, moreover, the ethnography indicates that rock writing sites “may have been used, during male initiations, to give the boys an idea of the appearance of the supernatural that they themselves were about to visit under the influence of toloache [Datura]” (2006:323; cf. Whitley and Carey 2016).
The archaeological record at Pinwheel Cave provides compelling evidence linking the Chumash rock writings with shamanistic rituals, practices and visionary experiences. That this site is located in a cave with an associated habitation deposit, in the San Andreas rift zone area, demonstrates its appropriateness for rock writing based on the multiple variables that contributed to the perception of this locale as supernaturally powerful.